# Genome Wide Identification and Characterization of Wheat *GH9* Genes Reveals Their Roles in Pollen Development and Anther Dehiscence

**DOI:** 10.3390/ijms23116324

**Published:** 2022-06-05

**Authors:** Liqing Luo, Jianfang Bai, Shaohua Yuan, Liping Guo, Zihan Liu, Haoyu Guo, Tianbao Zhang, Wenjing Duan, Yanmei Li, Changping Zhao, Xiyue Song, Liping Zhang

**Affiliations:** 1College of Agronomy, Northwest A&F University, Yangling 712100, China; luoliqing@nwafu.edu.cn (L.L.); 13683264256@163.com (L.G.); zhangtianbao8@126.com (T.Z.); duanwenjings@163.com (W.D.); lymei1013@163.com (Y.L.); 2Key Laboratory of Molecular Genetics in Hybrid Wheat, Beijing Academy of Agriculture and Forestry Science Research Institute, Beijing 100097, China; baijianfang131@163.com (J.B.); keaidehuahua0830@126.com (S.Y.); liuzihan643@163.com (Z.L.); guohaoyughy@126.com (H.G.); cp_zhao@vip.sohu.com (C.Z.)

**Keywords:** anther dehiscence, cellulose, GH9, phytohormone crosstalk, pollen sterility, wheat

## Abstract

Glycoside hydrolase family 9 (GH9) is a key member of the hydrolase family in the process of cellulose synthesis and hydrolysis, playing important roles in plant growth and development. In this study, we investigated the phenotypic characteristics and gene expression involved in pollen fertility conversion and anther dehiscence from a genomewide level. In total, 74 wheat *GH9* genes (*TaGH9s*) were identified, which were classified into Class A, Class B and Class C and unevenly distributed on chromosomes. We also investigated the gene duplication and reveled that fragments and tandem repeats contributed to the amplification of *TaGH9s*. *TaGH9s* had abundant hormone-responsive elements and light-responsive elements, involving JA–ABA crosstalk to regulate anther development. Ten *TaGH9s*, which highly expressed stamen tissue, were selected to further validate their function in pollen fertility conversion and anther dehiscence. Based on the cell phenotype and the results of the scanning electron microscope at the anther dehiscence period, we found that seven *TaGH9s* may target miRNAs, including some known miRNAs (miR164 and miR398), regulate the level of cellulose by light and phytohormone and play important roles in pollen fertility and anther dehiscence. Finally, we proposed a hypothesis model to reveal the regulation pathway of *TaGH9* on fertility conversion and anther dehiscence. Our study provides valuable insights into the GH9 family in explaining the male sterility mechanism of the wheat photo-thermo-sensitive genetic male sterile (PTGMS) line and generates useful male sterile resources for improving wheat hybrid breeding.

## 1. Introduction

Glycoside hydrolase family 9 (GH9) encodes a cellulase gene consisting of endo-β-1,4-glucanase (EGase) [1], which could hydrolyze soluble cellulose into reducing oligosaccharides. In addition, it was also shown that the GH9 family could hydrolyze cellulose-related polysaccharides such as carboxymethyl (CM) cellulose, phosphate-swollen cellulose and 1,3-β-1,4-β-glucan [2]. EGases have been studied in many species, such as eukaryotes and prokaryotes [3,4]. Based on their binding sites, the GH9 family can be divided into three subfamilies: class A (GH9A), class B (GH9B) and class C (GH9C) [5]. It has been reported that the members of GH9A and GH9B are involved in cell elongation and fruit maturation and abscission. GH9C has a cellulose-binding structural domain localized on the plasma membrane carbohydrate binding modules (CBM49). All these *GH9s* have been reported to be associated with cellulose biosynthesis [6]. 

Currently, there are many studies on the *GH9* family genes in plants, including *Arabidopsis* [7], rice [8], poplar [9], cotton [10] and strawberry [11]. In previous studies, *GH9* genes were found to be involved in many plant growth processes such as cell elongation, anther dehiscence, pollen tube growth, branch node abscission and fruit ripening [12,13,14]. In tomatoes, *Cel5* gene, a member of endo-β-1,4-D glucanase, is involved in pistil, pedicel and leaf abscission as well as fruit ripening [15]. In oranges, the increased polygalacturonase (PG) activity could induce the abscission of calyx abscission [16]. In *Arabidopsis*, plasma membrane-bound endo-1,4-β-D-glucanase play important roles in normal cell wall growth and cell elongation [17]. In addition, a defect of the *OsGLU1* gene leads to reduced cellulose levels in rice, which also affects rice cell elongation [18,19]. 

Anther normal development and dehiscence are key factors for plant fertility. Thus, the normal development of anther (the container of pollen), especially the anther wall, is the key to pollen development. Anther endothecium is composed of cellulose, pectin and endothecium wall proteins [20]. It was demonstrated that cellulases, hemicellulases and pectinases, such as PGs, β-1,4-glycosidases, and extensin, were involved in plant organ abscission and dehiscence, since they could directly participate in the modification and degradation of plant cell walls [21,22]. Cellulose is a component of the pollen wall and is necessary for the development and maturation of microspores [23]. Thickness is not uniform in a cellulose-deficient pollen wall, causing abnormal pollen wall growth. In addition, the results show that the glycoside hydrolase family genes are necessary for pollen wall growth in late anther development, involving anther dehiscence [24]. After periclinal division, the anther primary parietal cell differentiates into epidermis, endothecium (fibrous layer), one layer of middle layer and amoeboid tapetum. When the anther matures, the inner tangential wall and radial wall of the endothecium develop fibular thickenings, and an amount of starch is degraded to sugar, changing the osmotic potential of the anthers, contributing to anther dehiscence and pollen release. For example, in *Arabidopsis*, some fibrous thickenings in trinucleate anthers appear on the walls of endothecium cells. Additionally, in the flowering stage, endothecium cells and connective tissue cells shrink after losing water, resulting in anther dehiscence [22]. 

Recently, studies have implied that *GH9* may be associated with stamen development and anther dehiscence. It has been found that the glycosyl hydrolase *AtGHL* is highly expressed in *Arabidopsis* stamens. The mutation of *AtGHL* results in stamen deletion, suggesting that the *AtGHL* gene may be associated with stamen growth [25]. The expression level of the glycosyl hydrolase superfamily gene *TaBG* was higher in the anther dehiscence stage in a sterile environment and was 2.8 times higher in indehiscent anthers than that in normal dehiscent anthers in wheat photo-thermo-sensitive genetic male sterile (PTGMS) line BS366. The results suggest that the increase of *TaBG* gene expression could make the soluble sugar content increased in anthers, increasing the osmotic potential and slowing down the dehydration activity, resulting in anther indehiscence [26]. 

Wheat PTGMS line BS366, an important female parent material of two-line hybrid wheat, was developed by the DH (double haploid) population of Jingnong 8121 × E8075-7; its fertility is controlled by light and temperature. The main characteristics of male sterility on BS366 are microspore male sterility and low seed-setting rate due to anther indehiscence [27]. *GH9* genes were differentially expressed under different fertile environments in BS366. However, few studies have reported how cellulose regulates anther development and is involved in male fertility and how cellulase genes such as *GH9* participate in the regulation pathway in these biological processes. In this study, the *GH9* family was characterized to reveal the regulation of *GH9* on pollen development and anther dehiscence in wheat PTGMS lines. Here, phylogenetic tree, gene structure, conserved motifs and chromosomal localization comprehensively by bioinformatics methods for *GH9* gene family were analyzed. In addition, the expression of *TaGH9* in BS366 was detected using qRT-PCR. miRNAs such as miR167 and miR172 are associated with anther development in the wheat PTGMS line [28]. We also predicted the interactive relationships between *GH9* and microRNAs to further understand the roles of the *GH9* family, and finally a network regulatory map was constructed. Our study provides a good theoretical basis for further understanding the *TaGH9* function and provides a new sight for research on the mechanism of fertility transition in wheat PTGMS lines. 

## 2. Results

### 2.1. Identification of the GH9 Family in Wheat

After removing structurally incomplete genes, 74 copies of the *GH9* gene, which belonged to 24 *TaGH9* genes, were obtained. The coding sequence (CDS) lengths and protein lengths are shown in Appendix A. The predicted molecular weight (MW) of each GH9 protein ranged from 44.45 kDa to 70.62 kDa, with corresponding isoelectric points (IP) from 4.84 to 9.34. The aliphatic index (AI) ranged from 64.35 to 83.47. The total hydrophilic average (GRAVY) ranged from −0.475 to −0.089, suggesting all TaGH9s are hydrophilic proteins. Subcellular localization predictions revealed that the GH9 protein acts in the cytoplasmic, golgi, mitochondrial, endoplasmic reticulum and nucleus. The results suggest that different TaGH9 proteins may have different biological functions (Appendix A). 

### 2.2. Phylogenetic Analysis of the GH9 Protein

Combining the identified *GH9* families in *Arabidopsis*, rice and poplar, a phylogenetic tree of wheat was constructed. As shown in Figure 1, 74 copies of *TaGH9* were classified into three major subclades, Class A, Class B and Class C with 8 (10.8%), 58 (78.4%) and 8 (10.8%) copies of *TaGH9*, respectively. The Class B subclade is the largest subclade in the GH9 of wheat, which was further divided into three subgroups, Class B1, Class B2 and Class B3, with the numbers 37 (50%), 16 (21.6%) and 5 (6.8%), respectively.

### 2.3. Chromosomal Localization of the TaGH9 Family and Synteny Analysis

To understand the relative positions of the *GH9* genes on wheat chromosomes, we tagged their relative positions on wheat chromosomes. A total of 74 copies of *TaGH9* were distributed on 20 chromosomes, except chromosome 3B. The distribution of genes on chromosomes was scattered and there was no cluster distribution (Appendix A).

The synteny analysis revealed that there were three *TaGH9s* occurring in tandem replications, including *TaGH9-1-A1*/*TaGH9-1-A2*/*TaGH9-1-D1*/*TaGH9-1-D2*/*TaGH9-1D3*/*TaGH9-1-D4*, *TaGH9-11-A1*/*TaGH9-11-A2*/*TaGH9-11-B1*/*TaGH9-11-B2* and *TaGH9-20-A1/TaGH9-20-A2/TaGH9-20-A3* (Appendix A). The remaining 63 copies were all fragmental replications, suggesting that fragmental replication was associated with the evolution of the wheat *GH9* family and the expansion of members. The findings indicated that fragmental replication may be the main driver of the wheat *GH9* gene family expansion, in line with previous research [29].

### 2.4. Structure Analysis and Motif Distribution of the TaGH9 Family

The gene structure of *TaGH9* genes was understood by analyzing the exon–intron distribution. As indicated in Appendix A (left), *TaGH9* contained two to seven exons and introns. It was found that copies of the same *TaGH9* genes had similar or identical intron–exon constructions. For example, *TaGH9-13* have the same intron–exon structures among their copies, while the copies of *TaGH9-19* and *TaGH9-9* in a different subclade display different structures.

We also investigated the motifs of *TaGH9* to further revel the role of *TaGH9* using the MEME tool. In this study, ten types of motifs were charactered. The results show that the number of conserved motifs for each TaGH9 protein is mostly between seven and ten. Here, we found that *TaGH9-6*, *TaGH9-9* and *TaGH9-10* have 9 motifs without motif 9 (Appendix A (right)). The results show that the exon–intron structures and motifs of different TaGH9 members are different. However, the exon–intron structures and motifs were similar or the same in the same *TaGH9* subfamily. This suggests that the functional divergence of different *TaGH9s* may be caused by different exon–intron and motifs.

### 2.5. Cis-Acting Elements in Promoter Have Valuable Information for Analysis of the TaGH9 Family Function

The *cis*-acting element plays a key role in gene expression as an essential component of the gene promoter region, which assists to understand how the gene responds to external stimulation [30]. As shown in Figure 2, 29 categories of *cis*-element were identified, including light response-related elements, plant growth and development-related elements, hormone-response-related elements and biotic/abiotic adversity stress-related elements (Figure 2 and Appendix A). G-box, a light-response-related element, GARE-motif and TC-rich elements with temperature-related TGACG-motif, a MeJA-response elements and ABRE (ABA responsive element), an abscisic acid-response-related element, were found in the *TaGH9* gene family. There are differences in the *cis*-acting elements between different *TaGH9s*. For example, *TaGH9-11* and *TaGH9-12* had more light-responsive elements and hormone-responsive elements than stress-related elements and growth and development-related elements (Figure 2). LTR (low-temperature response) elements concerned with low-temperature treatment were enriched in *TaGH9-11*, suggesting that the *TaGH9* family has different roles since it owns different *cis*-acting elements. 

### 2.6. TaGH9 Genes Were Involved in Stamens Development

The gene expression of *TaGH9* in root, stem, leaf, spike and seed at the seedling, nutritional and reproductive stages were investigated using the most recent wheat transcriptome data. Available expression data are presented in Appendix A. The results show that most *TaGH9s* with stage-specific expression were differentially expressed in varied tissues. For example, *TaGH9-16* is highly expressed in the root and stem. 

Additional validation of the accuracy of *TaGH9* in various tissues by qRT-PCR analysis was also performed on 24 *TaGH9* genes to further characterize the accuracy of the *TaGH9* in selected tissues. In accordance with transcriptome data, these *TaGH9s* differentially expressed in all five tissues (Figure 3 and Appendix A). Interestingly, ten genes (*TaGH9-2, TaGH9-4, TaGH9-7, TaGH9-9, TaGH9-11, TaGH9-15, TaGH9-16, TaGH9-19, TaGH9-21* and *TaGH9-22*) were highly expressed in the stamens (Figure 3). 

### 2.7. TaGH9 Involving Pollen Development and Pollen Fertility Conversion

In our previous study, we found that *TaGH9s* were involved in pollen fertility conversion [31]. To investigate the roles of the *TaGH9* family on anther and pollen development in BS366, pollen and anther morphology at critical stages of pollen development stage were observed. It was found that there was malformation when anther and pollen in BS366 were cultivated under sterile environment. As shown in Figure 4, there was no obvious difference between morphology of anther at critical stages of pollen growth (stages 6–10). Under fertile condition, the anther could dehisce at stage 14 and be fertilized at stage 15 [32]. However, under sterile condition, anther indehiscence was showed at stage 14, then few or no pollen released from the anther, leading to unsuccessful fertilization in BS366 (Figure 4). 

To further understand the sterile characteristics of BS366, the pollen morphology under different fertility conditions was investigated. There was no difference between the microspore mother cell (stage 6) of fertile and sterile conditions. Compared with pollen under fertile condition, the cell plate at dyad (stage 7) under sterile condition cannot be properly formed in telophase I (Figure 5A), resulting in an abnormal tetrad (stage 8). Moreover, we found no significant difference in stage 9. At the late uninucleate stage (stage 10), the starch content was decreased, and no generative or vegetative cells were observed. The expressions of these ten *TaGH9s* in different pollen development stages were also investigated. The results show that *TaGH9-7*, *TaGH9-15* and *TaGH9-19* genes highly expressed under the sterile environment during the critical stages of pollen fertility conversion (Figure 5B).

### 2.8. TaGH9 Involving Anther Dehiscence

Anther dehiscence is an important characteristic for male sterility of the BS366. To understand the roles of *TaGH9*, the expression of these genes was tested during anther dehiscence stages. It was found that *TaGH9-2*, *TaGH9-4* and *TaGH9-7* had lower expression under the sterile environment, while *TaGH9-9* and *TaGH9-15* had higher expression (Figure 6A). 

We also measured the cellulose content of anthers during the anthesis period, including stage 13–15 (Figure 6B). The results show that the cellulose content of anthers was increasing during the anthesis period under fertile condition, while cellulose content was no longer increasing with flowering the four days before anthesis, suggesting that cellulose plays an important role in anther dehiscence. These results are highly consistent with the dynamic changes of *TaGH9-2*, *TaGH9-4* and *TaGH9-7* genes during the anther dehiscence period (Figure 6A). 

To characterize the anther dehiscence, anther wall was observed by SEM (scanning electron microscope) (Figure 7). The results indicate that almost no cracking showed in anther, and the middle layer was smooth without wrinkles at the cracking of anther under sterile condition. However, anther could crack, and the middle layer had more wrinkles at the cracking of anther under fertile condition. We also found there were more ubisch bodies in the inner anther wall under fertile condition than that under sterile condition.

### 2.9. MicroRNA Targeting Prediction of TaGH9 and Expression Verification

miRNAs could interact with genes and play an essential function in plant growth and development [33]. In our earlier study, we found that miRNAs target genes, regulating male sterility in the PTGMS line BS366. To further understand the function of *TaGH9* in the regulation of fertility, seven pollen and/or anther development-related *TaGH9s* were selected to evaluate the binding association with miRNAs. The results indicate that these seven genes harbored targets for ten different miRNAs (Figure 7 and Appendix A). According to the discovery, we found that tae-miR5384-3p, tae-miR96574-5p, tae-miR9661-5p, tae-miR9676-5p and tae-miR398 potentially target *TaGH9-15-A*. It has been proved that miR398 could directly target Cu/Zn superoxide dismutase genes, which was associated with antioxidant stress [34]. In rice, miR398 may also interact with the expansion precursor protein gene *(EXPB)*, involved in rice pollen germination [35]. In our previous study, the EXPB gene family could also interact with tae-aly-miRNA398a-5p/tae-blo-miRNA398a-5p, highly expressed in anthers under sterile environments, playing important roles in fertility regulation [28]. In the present study, tea-miR398 also may target *TaGH9-2-A/B/D*. In addition, other known miRNAs such as tae-miR164 and tae-531 were predicted to target with *TaGH9*.

To verify the relationship between miRNAs and *TaGH9*, qRT-PCR was performed. The results show that *TaGH9-9*, *TaGH9-16*, *TaGH9-15* and *TaGH9-19* were targeted with tae-miR164, tae-miR9661-5p, tae-miR5384-5p and tae-miR9676-5p, respectively, involved in pollen fertility conversion (Figure 8B). In addition, *TaGH9-2*, *TaGH9-4*, *TaGH9-15* and *TaGH9-9* were targeted with tae-miR398, tae-miR9661-5p, tae-miR398 and tae-miR164, respectively, involving anther dehiscence (Figure 8C). Meanwhile, the opposite expression patterns were found between miRNAs and the corresponding *TaGH9s* (Figure 8C).

## 3. Discussion

Glycoside hydrolase family 9 (GH9) was previously reported as cellulase family E, whose members are spatiotemporal expression differences in plant growth and development and play various roles by encoding various hydrolysis-type genes and binding different substrates [5]. In recent years, *GH9* have been identified from many plants such as rice (*Oryza sativa* L.), sorghum (*Sorghum bicolor* (L.) Moench), maize (*Zea mays* L.), short-stalked grass (*Brachypodium distachyon* (L.) Beauv.), *Arabidopsis thaliana*, poplar (*Populus*) and cotton (*Gossypium*. L). In our previous study, it was found that *GH9* may be involved in male sterility by RNA-seq [31]. *GH9* had differential alternative splicing between fertile and sterile condition, involved in fertility conversion. However, there are few reports regarding how *GH9* works in pollen and anther development in crop plants. A total of 24 wheat *GH9* with 74 copies were isolated in this study, to reveal the function of *GH9* in wheat pollen and anther development (Figure 1 and Appendix A). 

### 3.1. TaGH9s Involving Phytohormone Crosstalk-Regulated Pollen Development

As molecular switches, *cis*-regulatory elements could respond to external stimuli, affecting plant growth by regulating gene transcription [36]. To further understand the functions of *TaGH9*, the *cis*-regulatory elements in promoter regions of *TaGH9* were analyzed. It was found that four major *cis*-regulatory elements, including light-responsive elements, stress-related elements, growth and development-related elements and hormone-responsive elements, were identified from *TaGH9* (Figure 2 and Appendix A). Hormone-responsive elements are enriched in these *TaGH9s*, especially MeJA response elements (TGACG-motif and CGTCA-motif) and abscisic acid response elements (ABRE element) are enriched in the *TaGH9* family (Figure 4). The latest research has informed us that GA and JA signaling are associated with anther dehiscence in BS366 [29,37]. Phytohormone crosstalk as a massive network has been known to be involved in plant growth. JA–GA crosstalk plays an essential role in stamen development and regulates male plant fertility [38,39]. Anther dehiscence involves cell abscission [40]. In our study, almost all *TaGH9s* have an ABRE element which is involved in the ABA signaling pathway, suggesting that ABA signaling is related to anther dehiscence besides JA signaling. JA and ABA signaling may have crosstalk in *TaGH9*, playing roles in pollen and anther development. The pollen of BS366 could sense light periods and low temperatures and cause fertility conversion [41]. We also found that nearly all *TaGH9s* contained G-boxes, which are involved in light signaling, suggesting *TaGH9* may respond to light and be related to pollen development. To clarify the functions of *TaGH9*, expression profiles in different tissues (Figure 3) were investigated. Ten highly expressed genes (including *TaGH9-2*, *TaGH9-4*, *TaGH9-7*, *TaGH9-9*, *TaGH9-11*, *TaGH9-15*, *TaGH9-16*, *TaGH9-19*, *TaGH9-21* and *TaGH9-22*) in the stamens were identified based on the results of expression profiles of *TaGH9s* in tissues. These genes also contained more light and hormone-responsive elements. 

### 3.2. TaGH9 May Be Involved in Anther Growth and Pollen Fertility Conversion

It is well known that the anther wall consists of five layers, including the epidermis, endothecium, middle layer (two layers) and tapetum [42]. Lipid and polysaccharide metabolism are required for development of the anther wall, since polysaccharide metabolism could stimulate formation and degradation of different anther wall layers [43]. The pollen tube wall is formed by the extension of the anther inner wall, which is mainly composed of pectin polymers, cellulose and hemicellulose, involved in polysaccharide metabolism [44]. The deposition of cellulose could be catalyzed by the cellulose synthase complex (CSC), which consists of multiple individual cellulose synthase (CESA) proteins, synthesizing at the plasma membrane via the cellulose synthase (CESA) complex [45]. It has been reported that the catalytic core of CSC consists of three types of CESA subunits, including CESA1, CESA3, and CESA6, as a deficiency in either component results in a reduction in cellulose and abnormal amounts of pollen, leading to male sterility, suggesting that cellulose synthesis is necessary for pollen development [46]. In rice, the mutation of *Os12BGlu38*, a member of GH1, showed shrunken and hollow pollen and deficient mutant pollen. In addition, no inner wall was observed in the pollen of *Os12BGlu38* mutants using transmission electron microscopy (TEM) [47]. Several *GH9* have been charactered in *Arabidopsis* and rice. *OsGH9B18*, *OsGH9B6/14* and *OsGH9C2* in rice, *At3g43860*, *AtGH9A3*, *AtGH9A4* and *AtGH9B5/11/14/17* in *Arabidopsis*, specifically expressed in stamen and pollen [8,48]. Genes with close homology may have similar roles. In our study, we found that *TaGH9-4/7/9* classified into Class C with *OsGH9C2* and *TaGH9-2/19/15/11* classified into Class B1 with *AtGH9B14*, *AtGH9A3* and *OsGH9B18*, suggesting that these *TaGH9* genes may have a similar function with genes in rice and Arabidopsis (Figure 1). *TaGH9-19* especially has very close homology with *AtGH9B14*, *AtGH9A3* and *OsGH9B18*. We also found that miR164 may target *TaGH9-9* genes (Figure 8). In *Arabidopsis*, miR164 cleaves ORE1, which is a transcription factor of NO APICAL MERISTEM (NAM), participating in age-related cell death [49]. A lack of *miR164-CUC1*/2 results in the defect in sepals and low fertility in plants [50,51]. In addition, miR164 is induced by UV-B radiation in maize [52]. In our previous study, we also found miR164 was down-regulated in the pollen at the meiosis stage in sterile BS366 [28]. These results suggest that *TaGH9-9* genes interacted with miR164, participating in pollen fertility conversion by responding to light (Figure 8). 

*LSSR1*, encoding a GH5 cellulase, expressed in the anther microsporogenesis stage of rice. It has been reported that *LSSR1* with an N terminal signal peptide is secretory protein stored in pollen grains and plays important roles during fertilization in rice. The results indicate that the process of fertilization was inhibited in the mutant *LSSR1*, resulting in reduced fertility in rice, indicating that *LSSR1* plays an essential role in male fertility in rice [53]. In our previous study, following the formation of the cell plate, pectin and xyloglucan gradually combined, excess membrane began to disappear, cellulose content increased and eventually cell walls formed [54,55,56]. During meiotic cytokinesis, cellulose accumulates at both the division place and external wall of the cell [57]. At the prophase I stage, callose deposition happened in the pollen mother cell (PMC), forming a thick callose cell wall until the end of meiosis cell cycle [58]. Cellulose deposition is required for spore development during meiosis since it provides the basis for later pollen cell wall formation [56]. It has been reported that lack of cellulose deposition during meiosis could cause male sterility [59]. In our study, the cell plate abnormally formed in stage 7 (dyad) and stage 8 (tetrad) (Figure 5A), and homologous chromosomes failing to pair together were found in BS366 under sterile condition. In addition, *TaGH9-7*, *TaGH9-15* and *TaGH9-19* were highly expressed in the pollen development stage, especially in stage 7, 8 and 10, which were the critical stages of fertility conversion, indicating that highly expressed *TaGH9-7*, *TaGH9-15* and *TaGH9-19* during pollen development may be one of the reasons for pollen abortion (Figure 5A,B).

### 3.3. TaGH9 May Be Involved in Anther Dehiscence

Anther dehiscence depends on the following processes: expansion of the endothelium, deposition of fibrous bands in the inner layer, degradation of the septal cells and cracking of the dehiscence [40]. It has been reported that the content of cellulase in anther increases significantly during the development of anthers and pollens. Cellulase may also be involved in wall morphogenesis such as tapetum wall and callose enzyme activity. The timing of the callose enzyme activity is critical for proper male gametophyte development and the wrong timing may lead to male sterility [60]. In *Arabidopsis*, the *KORRIGAN* allele of *GH9A-rsw2-1* mutants showed that reductions in cellulose production, constitutively slow root growth and enhanced temperature-sensitive responses shortened stamen filaments and impaired anther dehiscence, while all characteristics were restored when the 8.5kb mucilage containing the *KORRIGAN* gene was transformed into the mutant *rsw2-1* [61]. In addition, overexpression of *PtrCel9A6*, a member of GH9, causes an increase in glycoside hydrolase activity, interferes with the accumulation of cellulose in epidermal cells and inhibita the thickening of the intra-anther wall, preventing normal anther dehiscence and pollen release, resulting in pollen grains remaining in their pollen sacs and ultimately leading to reduced fertility [62]. Moreover, the dehydration of anther wall cells also contributes to anther dehiscence. The cells of the anther wall dehydrate due to the decrease in water potential in the loculus. During the process of anther dehiscence, water content in anther cells decreased by transpiration, meanwhile, the transformation of starch into sugar increased the osmotic potential of anther tissue, promoting the success of the anther dehiscence process [63]. In tobacco, the defect of PIP2-like water channel proteins causes blocking of the dehydration process, resulting in abnormal anthers dehiscence [64]. Furthermore, in *Arabidopsis,* the H^+^-sucrose transporter AtSUC1, observed in the anther septum, could increase the osmotic potential of the septum and cause dehydration in the anther [65]. It was also reported that the concentration levels of flavonoid transport factor *FFT* in anther are involved in anther dehydration in *Arabidopsis* [66]. In our study, compared with anther dehiscence under fertile condition, the middle layer of anther was smooth, and few wrinkles were observed in indehiscent anther under sterile condition (Figure 7). The content of cellulose in anther under sterile condition was not increase with flowering, while it increased under fertile condition. In addition, highly expressed *TaGH9-9* and *TaGH9-15* at the anther dehiscence stage under sterile condition increased the osmotic potential of anthers wall cells, causing anther to fail to crack. Meanwhile, the down regulation of *TaGH9-4* and *TaGH9-7* may reduce the degradation rate of cellulose and failure to degrade septum, resulting in anther indehiscence.

In the *myb26* mutant of *Arabidopsis*, there is no radial expansion and secondary thickening of the wall cells within the anther chamber, resulting in the inability of the anther dehiscence, causing male sterility [67]. In wheat, it has been reported that miR398 targets the *EXPB* gene, up-regulating sterile anther involving pollen fertility conversion and pollen germination ability [28,68]. MiR398 could negatively interact with *TaGH9-2* and *TaGH9-15* at the anther dehiscence stage, indicating that these two genes may regulate the expansion of the cell wall and be involved in anther dehiscence (Figure 8 C).

### 3.4. A Putative Pathway of TaGH9 Regulating Pollen Fertility Conversion and Anther Dehiscence in Wheat

In this research, we formulated a possible regulatory pathway of *TaGH9* in pollen development and anther dehiscence (Figure 9). After stimulation by external environment with short day and low temperature at meiosis stage, wheat PTGMS line BS366 shows abnormal growth in pollen and anther. For pollen development, tae-miR164, tae-miR9661-5p, tae-miR5384-3p and tae-miR9676-5p showed competitive expression with target *TaGH9s* under sterile conditions. These *TaGH9s* were also involved in phytohormone signaling, such as JA–ABA crosstalk, contributing to the meiosis abnormalities and resulting in male sterility. Moreover, *TaGH9* is associated with anther dehiscence. In this study, we found that tae-miR164, tae-miR398 and tae-miR9661-5p targeted *TaGH9-9*, *TaGH9-15*, *TaGH9-2* and *TaGH9-4*, respectively, involving the cellulose content of the anther wall. The reduction in cellulose content in the anther wall resulted in anther indehiscence. Collectively, these findings suggested that *TaGH9s* might have a significant effect in male sterility in wheat PTGMS line BS366.

## 4. Materials and Methods

### 4.1. Plant Material, Growing Conditions and Sample Collection

In this study, the wheat photo-thermo-sensitive genetic male sterile (PTGMS) line BS366 (winter wheat) was used. Growing conditions: The seeds were planted in October 2020 in the experimental fields in Beijing (N 39°54′, E 116°18′) and Dengzhou (N 32°22′, E 111°37′), China, respectively, and managed conventionally.

Samples for fertility and anther dehiscence analysis were collected using the following equation reported by [26,37].

For tissue-specific expression analysis, the root, stem, leaf, petal, pistil, stamen and glume of wheat at the meiosis stage were collected.

For anther samples, eight different anther developmental periods in fertile and sterile environments, including microspore mother cells (Stage6), meiotic dyad (Stage7), meiotic tetrad (Stage8), early uninucleate (Stage9) and late uninucleate (Stage10), bicompartmental stage (Stage13), dehiscence stage (Stage14) and recession stage (Stage15), were collected (Figure 1).

### 4.2. Phenotypic Characterization of BS366

Anthers at different stages from fertile and sterile conditions were photographed with a ZEISS SteREO Discovery.V20 dissecting microscope (ZEISS, Jena, Germany). To further analyze pollen fertility conversion, the anthers of BS366 from meiosis to the mature pollen stage under both fertile and sterile conditions were collected and fixed in FAA solution (formaldehyde: glacial acetic acid: 50% ethanol = 5: 5: 9). The pollen grains were released from anthers by tweezers and dyed with improved carbol fuchsin solution [62] Photographs of microspores and pollen were obtained using an LEICA DM6000B microscope (LEICA, Wetzlar, Germany). For scanning electron microscope (SEM) analysis, anthers at the dehiscence stage were collected, fixed in 2.5% glutaraldehyde, dehydrated, air dried in silica, coated with gold–platinum in a sputter coater and finally examined by SEM (Phenom LE, Wetzlar, Germany) [69].

### 4.3. Data Sources for the GH9 Gene Family

The protein sequences of the glycoside hydrolase family 9 (GH9, EC3.2.1) of wheat were downloaded from the Ensembl Plant database (https://Plants.ensembl.org/Triticum_aestivum/Info/Index, accessed on 10 January 2022) and the Pfam database (https://Pfam.xfam.org/family, accessed on 10 January 2022). The GH9 protein data of Arabidopsis thaliana (At), Oryza sativa (Os) and poplar were downloaded from TAIR (http://www.Arabidopsis.org/index.jsp, accessed on 15 January 2022), RAP (https://rapdb.dna.affrc.go.jp/, accessed on 15 January 2022) and PIR (http://www.populus.db.umu.se, accessed on 15 January 2022).

### 4.4. Identification of Wheat GH9 Family Members

The Ex PASy-ProtParam online tool (https://web.expasy.org/protparam/, accessed on 13 February 2022) was used to analyze the basic physicochemical properties of the final obtained wheat GH9 protein, including gene CDS, amino acid number (AA), relative molecular weight (MW), isoelectric point (PI), total mean hydrophilicity value (GRAVY) and aliphatic index (AI). The subcellular localization of the wheat GH9 protein was predicted using the target P 1.1 Server (http://www.cbs.dtu.dk/services/Target~P/, accessed on 13 February 2022). GH9 subgenomic copies were named A, B and D according to their position on the chromosome.

### 4.5. Chromosomal Locations and Synteny Analysis

Positional information of *TaGH9* genes were collected based on the genome annotation information. The synteny analysis of *TaGH9* genes was performed by using the whole genome synteny block data, which was available in the Plant Genome Duplication Database (http://chibba.agtec.uga.edu/, accessed on 20 February 2022) [70]. The visualization of chromosomal locations of the *TaGH9* family was carried out by using Circos-0.69 (http://circos.ca/, accessed on 20 February 2022) [71].

### 4.6. Multiple Sequence Alignment and Phylogenetic Tree Construction

Multiple alignment of GH9 protein sequences was performed by using DNAMAN (ver. 6.0, LynnonBiosoft, CA, USA) with default parameters. The phylogenetic tree was constructed by using MEGA6.0 [49] with the neighbor-joining (NJ) method (1000 bootstrap trials and the Poisson model) according to the results of multiple alignment.

### 4.7. Structural Analysis of TaGH9 Genes and Proteins

Gene structure information of the TaGH9 family members were visualized using the Gene Structure Display Server (GSDS) tool [72] (http://gsds.cbi.pku.edu.cn/, accessed on 25 February 2022) by comparing the predicted GH9 coding sequences with their corresponding genomic sequences.

### 4.8. Analysis of cis-Regulatory Elements of GH9 Genes

To identify the cis-elements in the promoter sequences of the GH9 family genes in wheat, 1.5-kb of genomic sequences upstream of the promoter regions of each GH9 genes was chosen and screened against the Plant CARE database [73] (http://bioinformatics.psb.ugent.be/webtools/plantcare/html/, accessed on 25 February 2022) to identify the cis-elements.

### 4.9. RNA-Seq Data Analysis and Gene Expression Heatmap

The RNA-seq data of six tissues (root, stem, leaf, pistil, stamen and glume) and three developmental periods (seedling, vegetative and reproductive) of Chinese spring wheat were downloaded and analyzed using expVIP (http://wheat-expression.com, accessed on 27 February 2022). The TPM (Transcripts Per Kilobase of exon model per million mapped reads) data of *GH9* genes were visualized with a heat map using Graphpad Prism 9 software.

### 4.10. Measurement of Cellulose Content in Anther

Fresh wheat anthers of stage 13, 14 and 15 were collected and ground into powder under liquid nitrogen. The anthrone reagent was used to determine the cellulose content after constructing a standard curve for the quantification of cellulose [74].

### 4.11. miRNA Targeting Prediction of the TaGH9 Family

miRNA could play an important role in plant development by targeting genes. To investigate whether the candidate *TaGH9* interacts with miRNA, the known and studied miRNA sequences of wheat were collected from miRbase and subjected to online psRNATarget. The interaction networks between the candidate *TaGH9* and miRNAs were visualized by using Cytoscape (version 3.7, CA, USA) based on predicted results.

### 4.12. Expression Analysis of TaGH9 and miRNAs

The total RNA of all samples was extracted using TRIzol reagent (Invitrogen, CA, USA) following the manufactures’ instructions, and approximately 1 µg RNA was reverse transcribed with PrimerScript 1st Strand cDNA synthesis kit (TaKaRa, Shiga, Japan) and miRcute Plus miRNA cDNA T (Tingen, Beijing, China) for genes and miRNAs, respectively, according to the manufacturer protocol. qRT-RCR assays for TaGH9 and miRNAs were conducted on an Eco Real-time PCR system (Illumina, CA, USA) using TB Green Premix Ex Taq (Tli RNaseH Plus) kit (TaKaRa, Shiga, Japan) and SYBR^®^ miRcute Plus miRNA PreMix (Tingen, Beijing, China), respectively, as described previously [28,75]. Wheat 18S gene served as the reference gene for TaGH9 analysis, and U6 was used as a reference gene for miRNA analysis. The relative expression levels were calculated according to the 2^–ΔΔCt^ method for relative expression with three biological replicates and three technical replicates [76]. The primers used in this study were designed using the Primer premier 5.0 (Primer, CAN, UK) program and are listed in Appendix A.

## 5. Conclusions

In this study, a comprehensive overview of the *TaGH9* gene family in wheat, including gene and protein structures, phylogenetic relationships and expression profiles, was provided. A total 74 copies of *TaGH9* were identified and classified into three major subclades, named Class A, Class B and Class C. The *TaGH9**s* in the same subfamily and subclades may own similar functions. There were abundant light-response-related elements and hormone-response-related elements in the promoter region that assist *TaGH9**s* to respond to light and hormones, regulating anther and pollen development. Ten *TaGH9**s* were highly expressed in stamens. Finally, seven *TaGH9s* may be associated with pollen development and anther dehiscence. A putative pathway of *TaGH9s* with targeted miRNA regulating to pollen fertility conversion and anther dehiscence was constructed. The potential function of *TaGH9* needs to be further verified using more molecular biological techniques and tools. Collectively, these findings establish a foundation for further exploration of *TaGH9* and provide novel insights into its biological functions.

## Figures and Tables

**Figure 1 ijms-23-06324-f001:**
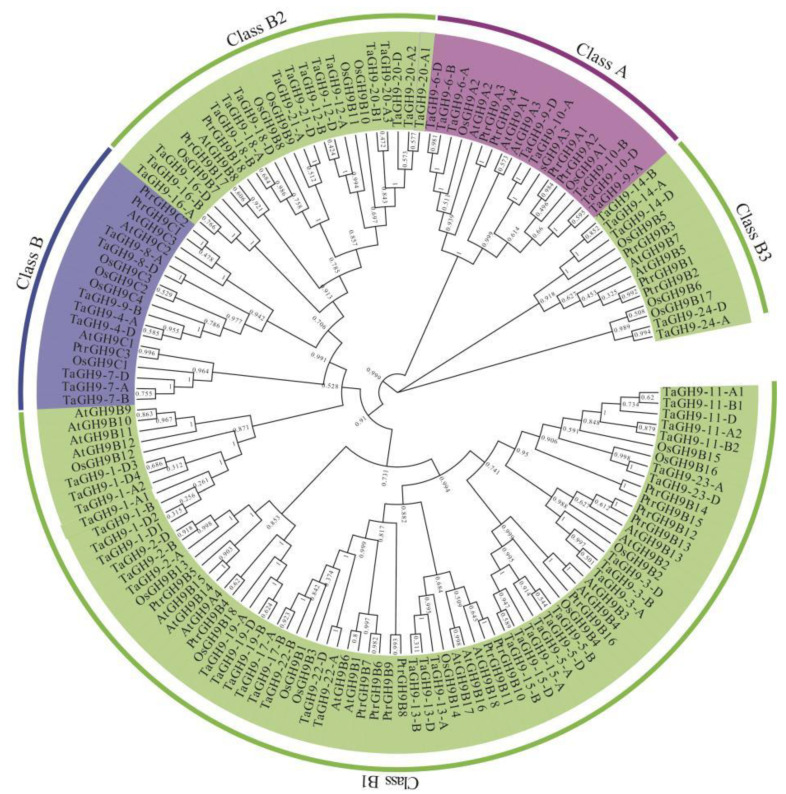
Phylogenetic trees of the GH9 proteins from wheat, *Arabidopsis*, poplar and rice. The complete amino acid sequences of 22 Arabidopsis, 24 rice and 24 poplar GH9 proteins were compared with DNAMAN, and the phylogenetic trees were constructed using MEGA 7.0 with 1-K bootstrap replicates using the maximum likelihood method. Different colors represent different subclades.

**Figure 2 ijms-23-06324-f002:**
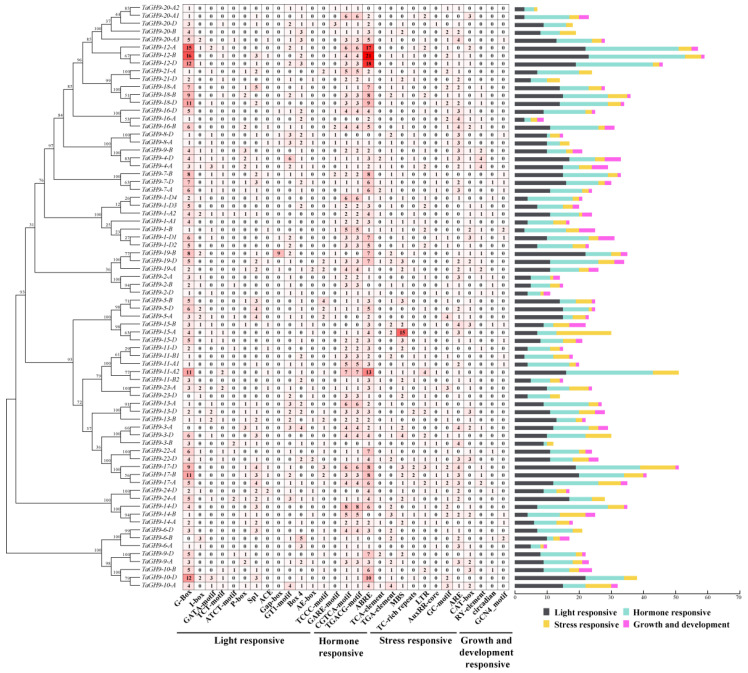
Analysis of cis-acting regulatory elements in the *TaGH9* promoter region. The various colors mean various elements. Heat map with gradual red means the number of each *cis*-acting regulatory element in the corresponding *TaGH9*.

**Figure 3 ijms-23-06324-f003:**
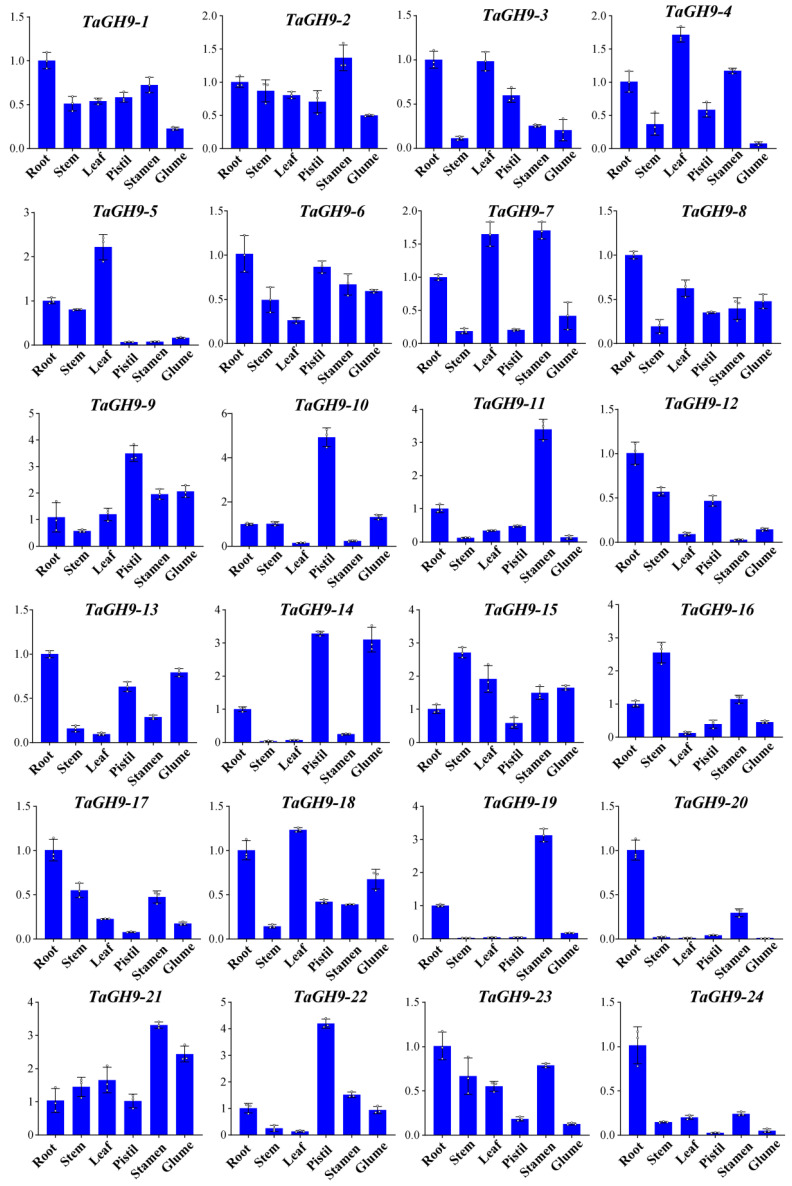
Expression of 24 genes in the roots, stems, leaves, pistils, stamens and glumes of BS366. Data were calculated by the 2^−ΔΔCt^ method for relative expression, with three biological and technical replicates.

**Figure 4 ijms-23-06324-f004:**
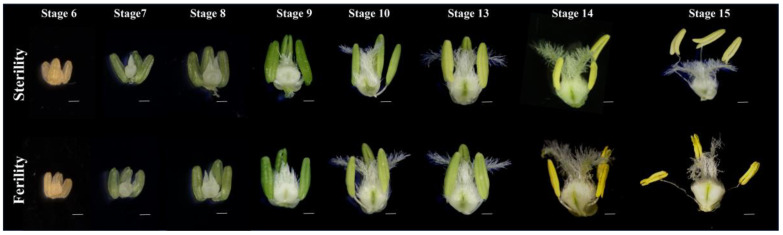
Phenotype comparison of BS366 anthers at critical pollen fertility conversion stages (stages 6–10) and anther dehiscence stages (stages 13–15) under sterile and fertile conditions. Bars = 100 μm.

**Figure 5 ijms-23-06324-f005:**
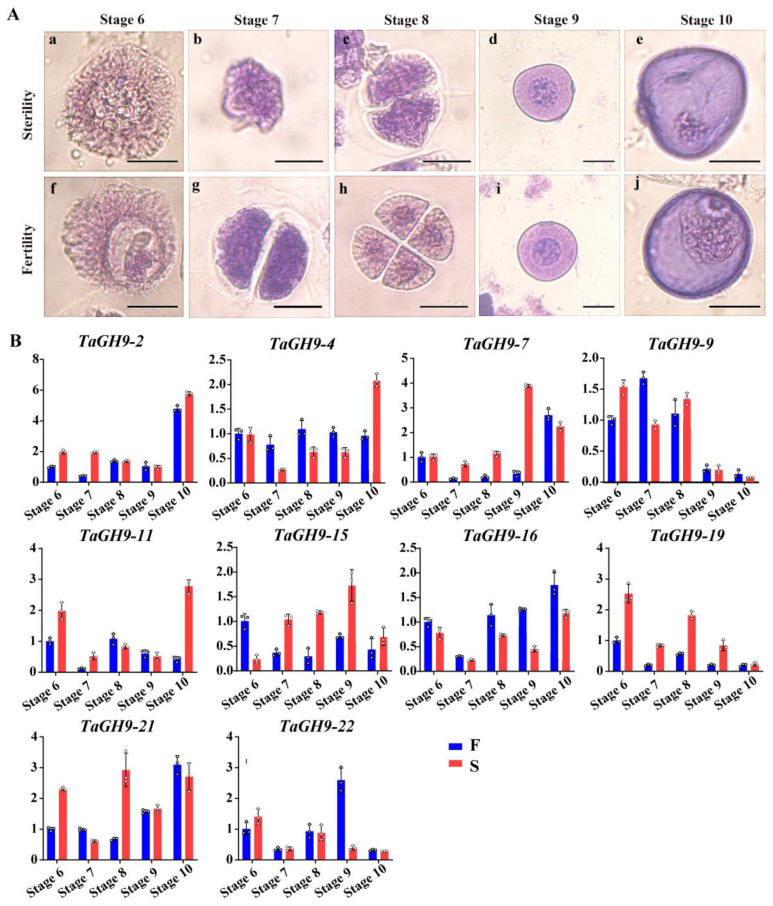
The microspores (**A**) and expression profiles (**B**) in the fertile and sterile BS366 of pollen fertility conversion stages (Stages 6–10). Micropores of sterile (**a**–**e**) and fertile BS366 pollen (**f**–**j**). Microspore mother cell (**a**,**f**), meiotic dyad (**b**,**g**), meiotic tetrad (**c**,**h**), early uninucleate stage (**d**,**i**) and late uninucleate stage (**e**,**j**). Bars = 20 μm. The data of expression profiles are from three biological replicates and three technical replicates, and error bars mean the standard error.

**Figure 6 ijms-23-06324-f006:**
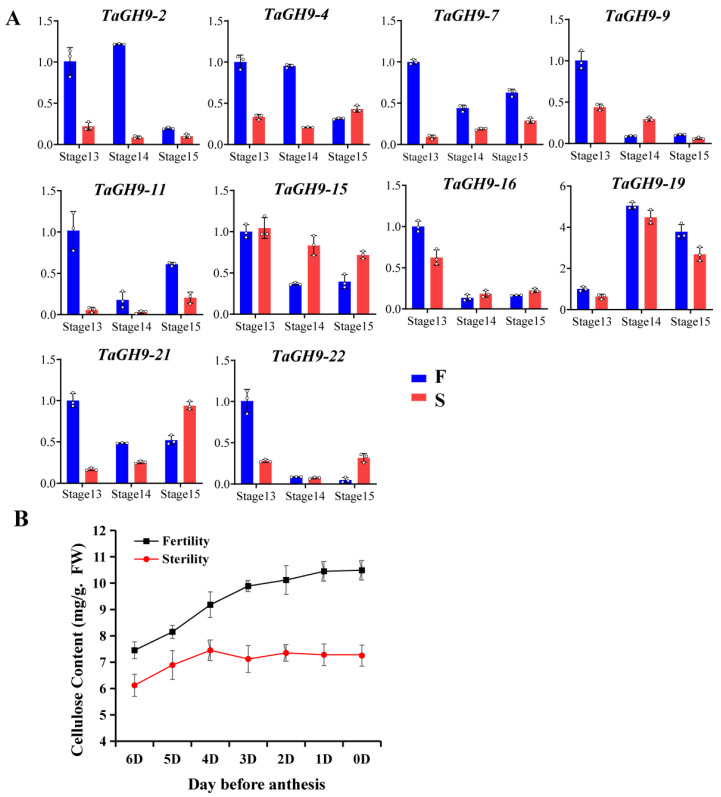
*TaGH9* involving anther dehiscence. (**A**) The expressions of 10 genes in anther dehiscence stages (Stages 13–15). Data were calculated by the 2^−ΔΔCt^ method for relative expression, with three biological replicates and technical replicates. Two-factor ANOVA was used (*p* < 0.01). (**B**) Cellulose content of anther during anther flowering. The data are from three biological replicates and three technical replicates, and error bars mean the standard error.

**Figure 7 ijms-23-06324-f007:**
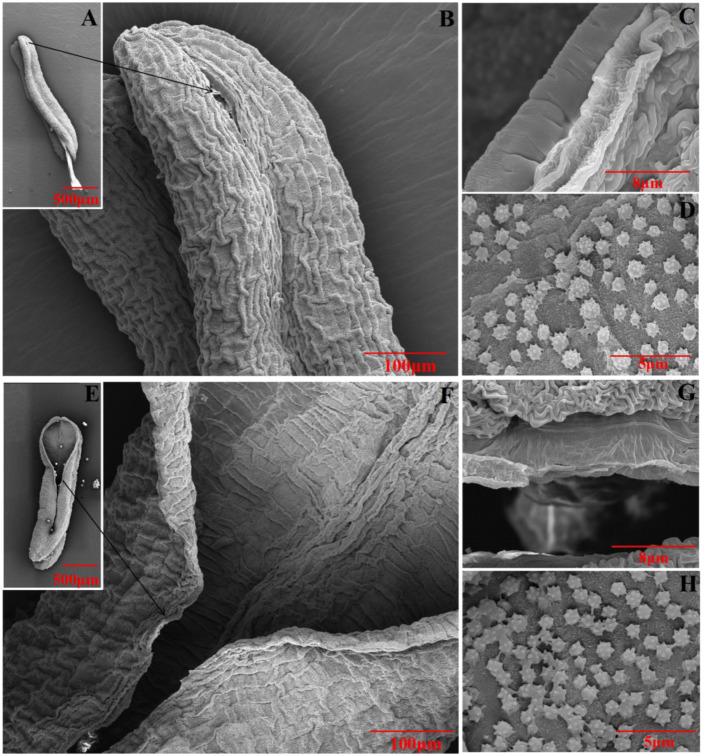
Observation of anther dehiscence under sterile condition (**A**–**D**) and fertile condition (**E**–**H**) by SEM. Corresponding bars were marked in photos.

**Figure 8 ijms-23-06324-f008:**
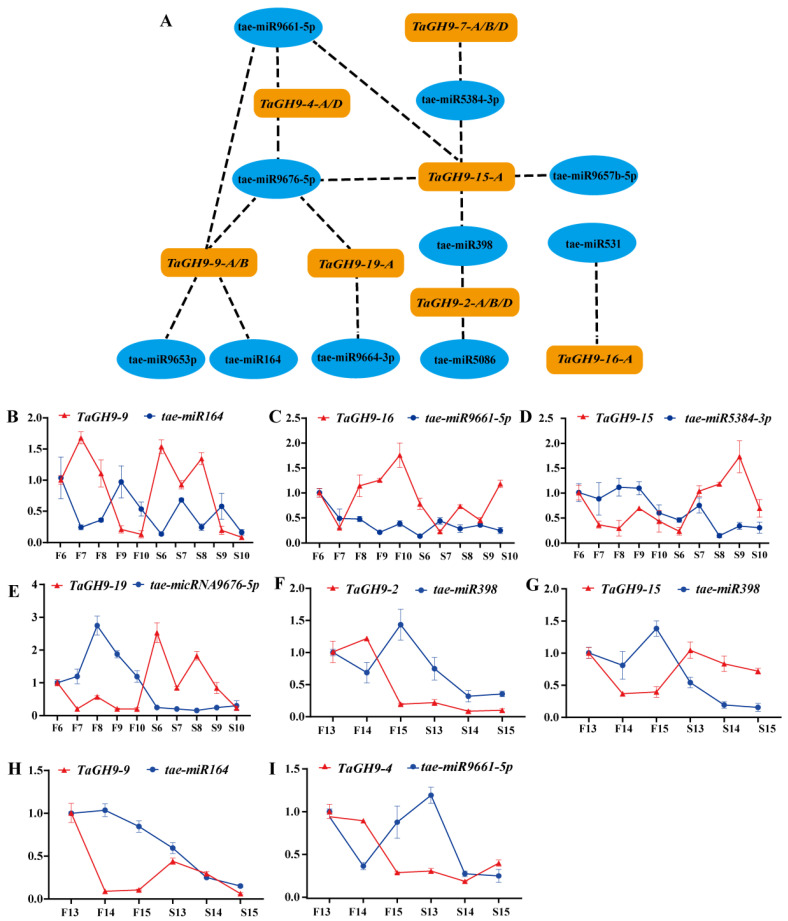
Network (**A**) and expression profiling relationship between miRNAs and their target *TaGH9* in pollen fertility conversion stages (**B**–**E**) and anther dehiscence stages (**F**–**I**). The data are from three biological replicates and three technical replicates, and error bars mean the standard error.

**Figure 9 ijms-23-06324-f009:**
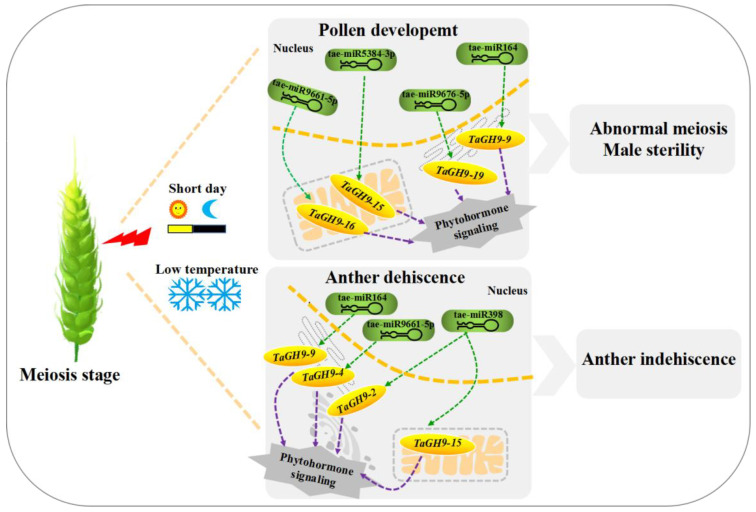
A putative pathway of *TaGH9* regulating to pollen fertility conversion and anther dehiscence in wheat. Organelles under orange ovals with gene names correspond to predicted subcellular locations for *TaGH9*.

## Data Availability

Not applicable.

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
