# Peer review of "Genome Wide Identification and Characterization of Wheat GH9 Genes Reveals Their Roles in Pollen Development and Anther Dehiscence"

_ijms, 2022, doi:10.3390/ijms23116324_

Round 1
Reviewer 1 Report
Manuscript "Genome-Wide Identification and Characterization of Wheat GH9 Genes Reveals Their Roles in Pollen Development and
Anther Dehiscence" by Liqing Luo, Jianfang Bai, Shaohua Yuan, Liping Guo, Zihan Liu, Haoyu Guo, Tianbao Zhang, Wenjing Duan, Yanmei Li, Changping Zhao, Xiyue Song and Liping Zhang presents data from an original study on the genetic aspects of anther formation and fertility. wheat pollen.
This aspect is of interest for practical breeding and experimental botany.
Although the concept proposed by the authors is not the only correct one in terms of explaining the phenomenon of anther opening features, it is still of interest to researchers working in this field. The disadvantage of the work is ignoring the reasons leading to impaired dehydration during opening, not related to dehydration, but related to the developmental features of the anther, which remained misunderstood despite the presence of microscopic analysis.
The quality of the scanning micrographs is low and it is desirable to improve it, since the pollen is not distinguishable.
I recommend adding a discussion of aspects of the causes of dehydration disorders.
The work is interesting and can be accepted for publication.
Author Response
Response to Reviewer 1 Comments
Point 1: Although the concept proposed by the authors is not the only correct one in terms of explaining the phenomenon of anther opening features, it is still of interest to researchers working in this field. The disadvantage of the work is ignoring the reasons leading to impaired dehydration during opening, not related to dehydration, but related to the developmental features of the anther, which remained misunderstood despite the presence of microscopic analysis.
The quality of the scanning micrographs is low and it is desirable to improve it, since the pollen is not distinguishable..
Response 1: Thank you for your good suggestions. The modified new figures have been replaced at corresponding position.
Point 2: I recommend adding a discussion of aspects of the causes of dehydration disorders.
Response 2: Thank you for your good suggestions. We added related discussion of aspects of the causes of dehydration disorders at corresponding position (section 3.3).
The details are as followed
“Moreover, dehydration of anther wall cells also contribute to anther dehiscence. Cells of anther wall dehydration, due to decrease of water potential in loculus. During the process of anther dehiscence, water content in anther cells decreased by transpiration, meanwhile, the transformation of starch into sugar increased the osmotic potential of anther tissue, promoting the successful of the anther dehiscence process [63]. In tobacco, defect of PIP2-like water channel proteins, causes blocking of the dehydration process, resulting in abnormal anthers dehiscence [64]. Furthermore, in Arabidopsis, the H+-sucrose transporter AtSUC1, observed in the anther septum, could increase the osmotic potential of the septum and caused dehydration in the anther [66]. It also reported that the concentration levels of flavonoid transport factor FFT in anther are involved in anther dehydration in Arabidopsis[67]”

Reviewer 2 Report
The mansucript ,,Genome-Wide Identification and Characterization of Wheat GH9 Genes Reveals Their Roles in Pollen Development and Anther Dehiscence,, contains interesting results and solves good topic.
The manuscript should be significantly improved or resubmitted. I recommend a huge major revision.
The structure of the manuscript must be improved as well as its interpretation. Go through the whole manuscript and improve its interpretation.
The abstract should be improved, you should include the backround, results and conclusions.
A list of the most important abbreviations could be given at the beginning of the manuscript. This would make it easier to find your way around the manuscript.
It would make sense if you described the materials and methods first and then the results and discussion.
Throughout the manuscript, the figures are very small and difficult to see. Improve the interpretation of some figures so that the descriptions are clearly visible in them.
,,In addition, mutation of OsGLU1 in rice results in cell elongation-blocked and decrease of cellulose content in cell wall [17]. Anther normal development and dehiscence are key factors for plant fertility.,, This issue has been addressed in great detail by this very important paper and therefore the authors are encouraged to add it here as a reference. https://www.sciencedirect.com/science/article/abs/pii/S0048969722011354
At the end of the introduction, state what makes this manuscript novel and innovative.
Improve the interpretation of chapters 2.1-2.6
Conclusions: Extend the conclusions with the most important findings. Please, indicate the possible risks of such research. Add your recommendations for future research.
References: Make sure the references are added correctly according to the journal's instructions.
Author Response
Response to Reviewer 2 Comments
Point 1: The abstract should be improved, you should include the background, results and conclusions.
Response 1: Thank you for your good suggestions. The related-contents have been modified at corresponding parts (Abstract).
Point 2: A list of the most important abbreviations could be given at the beginning of the manuscript. This would make it easier to find your way around the manuscript.
Response 2: Thank you for your good suggestions. The abbreviations have been added at corresponding parts (At the end of manuscript)
Point 3: It would make sense if you described the materials and methods first and then the results and discussion.
Response 3: Thank you very much for your careful reading of the manuscript. As you said, it makes sense to describe the materials and methods first. However, given the format requirements of the journal, we have to arranged the results and discussion before the materials and methods.
Point 4: Throughout the manuscript, the figures are very small and difficult to see. Improve the interpretation of some figures so that the descriptions are clearly visible in them.
Response 4: Thank you for your good suggestions. The figures have been modified at corresponding parts.
Point 5: In addition, mutation of OsGLU1 in rice results in cell elongation-blocked and decrease of cellulose content in cell wall [17]. Anther normal development and dehiscence are key factors for plant fertility.,, This issue has been addressed in great detail by this very important paper and therefore the authors are encouraged to add it here as a reference. https://www.sciencedirect.com/science/article/abs/pii/S0048969722011354
Response 5: Thank you for your good suggestions. The related-contents have been modified and cited the article at corresponding parts (reference 19).
Point 6: At the end of the introduction, state what makes this manuscript novel and innovative.
Response 6: Thank you for your good suggestions. We strongly agree with your suggestion. The related-contents have been modified at corresponding parts (Fifth paragraph of Introduction).
Point 7: Improve the interpretation of chapters 2.1-2.6.
Response 7: Thank you for your good suggestions. The related-contents have been modified at corresponding parts.
Point 8: Conclusions: Extend the conclusions with the most important findings. Please, indicate the possible risks of such research. Add your recommendations for future research.
Response 8: Thank you for your good suggestions. The related-contents have been modified at corresponding parts.
Point 9: References: Make sure the references are added correctly according to the journal's instructions.
Response 9: Thank you for your good suggestions. The references have been modified at corresponding parts according to the journal's instructions.

Round 2
Reviewer 2 Report
The manuscript ,,Genome-Wide Identification and Characterization of Wheat GH9 Genes Reveals Their Roles in Pollen Development and Anther Dehiscence,, has been significantly improved. I recomend publish this manuscript.